# Cultural adaptation, translation and validation of the conflict in adolescence dating relationship inventory (CADRI) in the Greek language

Esperanza Barroso-Corroto[1,2], Juan Manuel Carmona-Torres[1,2,*],
José Alberto Laredo-Aguilera[1,2], Ángel López-Fernández-Roldan[1,3],
Carlos Navarrete-Tejero[1], Evangelia Kartsoni[4], Antonios Christodoulakis[4],
Athina Patelarou[4], Michail Zografakis-Sfakianakis[4]

**1** Facultad de Fisioterapia y Enfermería, Universidad de Castilla-La Mancha, Toledo, Spain, **2** Grupo de Investigación Multidisciplinar en Cuidados, Universidad de Castilla-La Mancha, Toledo, Spain, **3** Hospital Universitario de Toledo, Toledo, Spain, **4** Department of Nursing, Faculty of Health Sciences, Hellenic Mediterranean University, Crete, Greece

* juanmanuel.carmona@uclm.es

## Abstract

### Background

Dating violence (DV) is a major public health problem with serious consequences for the young population. The Conflict in Adolescent Dating Relationship Inventory (CADRI) is the most internationally used tool to measure DV. However, no tool has been translated, culturally adapted, or validated in the Greek context to assess the prevalence of DV in the Greek population.

### Aim

To culturally adapt, validate, and translate the Conflict in Adolescent Dating Relationship Inventory (CADRI) in the Greek language to ensure its reliability and cultural and linguistic validity in the Greek population.

### Methods

A cross-cultural adaptation process and cross-sectional study were carried out. There were two phases in the research. First, face validity was assessed after the first phase of translation, which included cross-cultural item adaptation, content validation and expert review of the instrument. In the second stage, the scale's psychometric qualities were assessed on a sample of nursing college students. A CADRI instrument was used to assess DV.

### Results

The final sample comprised a total of 177 university students. The internal consistency and reliability were good (>0.7), with a Cronbach's alpha of 0.889 for the violence

provided the original author and source are credited.

**Data availability statement:** We have added the data from the article as supplementary information. Therefore, all relevant data and minimal data set are within the paper and its Supporting Information files. The results of this research are published in this article. Further information can be obtained from the authors upon reasonable request.

**Funding:** This research was funded by a grant from the European Regional Development Fund (ERDF) [Fondo Europeo de Desarrollo Regional (FEDER), (DOCM 27/01/2021)]. Esperanza Barroso-Corroto is supported by a grant (SBPLY/23/180502/000002) from the Junta de Comunidades de Castilla-La Mancha (Spain) and cofinanced by the European Social Fund Plus (ESF+ 2021-2027) Program.

**Competing interests:** The authors declare no conflict of interest.

perpetrated subscale and 0.925 for the violence suffered subscale, indicating strong correlations between the total item scores. After exploratory factor analysis was performed, the structure obtained was similar to the original structure. Furthermore, 88.7% of university students perpetrated DV in the last year on their partners, and 90.7% suffered DV, with verbal and emotional violence the most common types.

## Conclusion

This study provides robust evidence of the validity and reliability of the CADRI for measuring DV in the Greek university population. Additionally, the high prevalence makes it necessary to create DV prevention plans and further research, especially among nursing students.

## Impact

Validation of the CADRI in Greece will increase the knowledge of DV in this population, which lacks validated instruments.

## Public contribution

The results obtained with the CADRI can be applied to the prevention of DV in young people.

## 1. Introduction and background

Dating violence (DV) is a type of intimate partner violence that occurs between young dating couples and includes physical violence, sexual violence, physiological aggression, and stalking [1].

In Europe, studies on DV have been carried out in several countries, including Spain, Italy, the UK, and Germany. According to a systematic review, the prevalence rates of DV in European adolescents [2] varied for psychological victimization, ranging from 5.9% to 95.5% for females and 5.6% to 94.5% for males; physical DV victimization, from 2.2% to 32.9% for females and 0.8% to 29.8% for males; and sexual DV victimization, from 4.8% to 41.0% for females and 2.4% to 39.0% for males. These data are consistent with those reported in other studies conducted mainly in North America and in other countries where this phenomenon occurs [3] and in those carried out on young adults [4].

Being involved in any pattern of violence has negative consequences, especially among young people [5]. DV is associated with anxiety, depression, suicidal thoughts, emotion dysregulation, attachment insecurities, low self-esteem, low emotional well-being, posttraumatic stress, and poor academic performance [6–9]. Additionally, alcohol consumption, marijuana use, exposure to domestic violence, and participation in vandalism are other factors associated with DV [10,11].

Most of the abuse in the young adult and adolescent population is bidirectional, involving both men and women as victims and perpetrators [12,13]. While there is a significant prevalence of abuse among young adults and adolescents, young adults are more likely to experience more severe acts and repercussions [14].

There have been almost no studies in Greece addressing DV. Kalaitzaki et al. [15] conducted a study to address the relationship between dating violence and family violence as part of a European project. Sexual violence has been studied in young Greeks by Krahé et al. [16] through a multicountry study and by Sakellari [17], who performed a qualitative study

of sexual harassment. Intimate partner violence (IPV) in the Greek adult population has rarely been studied [18,19]. Although few studies exist, the Conflict Tactics Scale (CTS2) has been used to assess the prevalence of IPV. Additionally, the researchers in these studies used careful back-translation without instrument validation. No instruments have been adapted, translated, or validated to measure DV or IPV in Greek, making it difficult to study DV in this population.

The Conflict in Adolescence Dating Relationship Inventory (CADRI) developed by Wolfe et al. [20] was designed to measure abusive behavior in adolescent couples. The questionnaire is composed of 35 items and two subscales (perpetration and victimization) and can differentiate between physical abuse, sexual abuse, threatening behavior, verbal or emotional abuse, and relational aggression. According to a systematic review [21], 29 instruments for measuring DV were evaluated, and the CADRI was the most suitable.

The CADRI has been translated, culturally adapted, and validated in Spanish [22], demonstrating validity and reliability. Following this, a cultural adaptation to the Colombian population [23] was also carried out, finding reliability indices similar to those of previous versions with adequate internal consistency. At the same time, reduced versions of 10 items [24] and 15 items [25] have been developed, which, although adequately valid, are less sensitive than the original versions for detecting cases of DV.

The CADRI has also been validated among the university population, first in the 19–21-year-old population [26] and then in the 19–25-year-old population [27], revealing adequate fit indices, significant saturations, and high internal consistency indices.

Additionally, the factor structure of the CADRI is consistent across gender, race, ethnicity, and time, indicating that its use is appropriate for subjects with different characteristics [28].

## 1.1.  Aim

As previously mentioned, the CADRI is the most recommended tool for measuring DV. Although the CADRI has been adapted to different countries and cultures, it is not adapted to the Greek population, and it would be advisable to perform this validation to study and analyze this problem in Greek settings. Therefore, the aim of this study was to culturally adapt, validate, and translate the CADRI in the Greek language to ensure its reliability and cultural and linguistic validity in the Greek population, specifically in young adults, who constitute the university population. This process will enable an accurate assessment of the prevalence, characteristics, and effects of DV among Greek youth, providing a robust and standardized tool for use in future research, prevention, and support interventions. Additionally, the second objective was to determine the prevalence of dating violence in the Greek university population.

## 2.  Methods

### 2.1.  Design

A cross-cultural adaptation process and cross-sectional study were carried out. The data were collected in June 2024; the start date was 06/11/2024, and the end date was 06/30/2024. There were two phases in the research. First, face validity was conducted after the first phase of translation, cross-cultural item adaptation, and content validation. In the second stage, the scale was tested on a sample of college students to assess its psychometric qualities (exploratory factor analysis and reliability analysis).

The study was reported based on the Consensus-based Standards for the Selection of Health Measurement Instruments (COSMIN) checklist [29,30].

## 2.2. Participants and sample size

The reference population was Hellenic Mediterranean University (HMU) nursing students who were active in 2023–24. The inclusion criteria were being under 25 years old and having or having had a dating partner during the last year.

The criteria for conducting a factor analysis were used to calculate the sample size for cultural adaptation/validation. These criteria involve 5–10 subjects for each item [29,30]; therefore, as the CADRI scale is composed of 35 items, a sample of at least 175 participants was needed.

Additionally, for the cross-sectional study sample to be representative of HMU students, the prevalence of DV (53%) in university students reported in the study by Barroso-Corroto et al. [31] was used. The population estimate was calculated using the GRANMO program. A sample size of 219 randomly selected subjects would be enough to estimate with a 95% confidence level and a precision of +/-5 participants, a population percentage of approximately 53%. A substitution rate of 20% is anticipated.

To reach the sample size, participants were recruited during academic classes to ensure as much participation as possible. A member of the research team asked the teachers for permission, gave the study information and requested the students' voluntary participation. Once they agreed to participate, the informed consent form was given to the students for their signature.

## 2.3. Measures

Sociodemographic variables such as sex, age, current dating status, sexual orientation, weight, height, number of dating couples, duration of relationship, and type of relationship were included.

A CADRI instrument was used to assess DV. The CADRI is a scale designed by Wolf et al. [20] to measure DV in adolescents and young adults. This instrument comprises 35 items divided into 2 subscales capable of differentiating between victims and perpetrators of DV; it assesses different dimensions of conflict, including verbal, physical, emotional, and sexual aggression behaviors, as well as conflict resolution strategies. This instrument also uses a 4-point Likert-type response scale with the following response options: never, seldom, sometimes, and often.

## 2.4. Procedure

After the original scale was obtained with the author's permission (Wolf et al.), a backward and forward translation method was used to develop a preliminary Greek version of the CADRI for the different items.

The text was comprehensively examined to conduct a back translation and cultural adaptation. Two bilingual independent translators translated the original English scale into the Greek language. The Greek version was then back-translated into English to ensure the accuracy of the translation.

The translation was analyzed by a group of HMU professors, who found semantic, conceptual, content, and criterion equivalence with the original instrument. A single version of the scale was agreed upon, with changes to the items to accommodate the defining elements of DV and culture in the Greek context and not just a mere translation of the items. The same Likert-type score was maintained for the Greek version, with four response options.

Facial validity was conducted to analyze the clarity, accuracy, and comprehension of the items agreed upon in the previous phase. The final version of the scale was administered via a self-administered Google Form questionnaire and distributed to a representative sample of Greek students during academic lectures.

Before disseminating the questionnaire, the professors agreed that collaboration and participation in the study were not detrimental to the participants' academic performance. Attempts were made to reduce sample bias by reaching out to as many different lectures as possible and providing clear information about the study, highlighting the importance of the participation of all of them.

## 2.5. Data analysis

Statistical analysis was carried out with IBM SPSS statistic version 29, licensed by the University of Castilla-La Mancha.

Cronbach's alpha and McDonald's Omega were calculated to assess the internal consistency of the scale as a whole and each of its subcomponents. A Cronbach's alpha and McDonald's omega above 0.70 indicated adequate internal consistency. A correlation between items and item totals was assessed to prove the relevance and usefulness of each item in the scale.

The construct validity of the scale was verified via exploratory factor analysis (EFA), Kaiser–Meyer–Olkin (KMO) and Bartlett's tests of sphericity were used for sample adequacy. The percentage of variance explained by each factor was measured, and a varimax rotation with Kaiser normalization was applied.

Student's t tests and analysis of variance (ANOVA) were conducted to investigate differences in scale scores according to demographic variables such as gender, age, and marital status, providing information on the sensitivity of the scale to relevant differences in the population.

Pearson correlations were used to explore relationships between dating violence scale scores, and a significance level of $p < 0.05$ was considered for all tests.

A 35-item Likert-type scale was used (25 items on violence and 10 on positive behaviors in relationships that are not addressed in this study), where a value of 0 corresponds to never, 1 is rarely (on one or two occasions), 2 is sometimes (understood as having happened between 3 or 5 times) and 3 implies that it happens frequently. The variables have been dichotomized as 0 for never and the rest as 1. Whether violence is present has been calculated, as in previous articles [31,32].

## 2.6. Ethics approval and consent to participate

The present study was approved by the Hellenic Mediterranean University Ethics Committee (Heraklion) with code No 12435/31-5-2024.

The information sheet was attached to the online questionnaire. Prior to answering the study questions, all the participants read and consented to participate in the study. If they did not agree to participate in the study by marking the affirmative answer, "Yes, I have read the information sheet and agree to participate in the study", they could not proceed to answer the online questionnaire.

## 3. Results

### 3.1. Sociodemographic characteristics

The sample comprised 228 university students, with a mean age of 21.42 years (SD ± 1.77): 40 men, 227 women and one person of undefined gender. Regarding the sexual orientation of the participants, 87.7% defined themselves as heterosexual, 4.4% as homosexual, 5.3% as bisexual and 2.2% claimed to have a different sexual orientation. Forty-four of the participants were not currently in or had not been in a dating relationship in the previous year; thus, they were excluded from the sample because they did not meet the inclusion criteria, as shown below (Fig 1). Among those who were dating or had dated, 15.8% were relationships with a group of

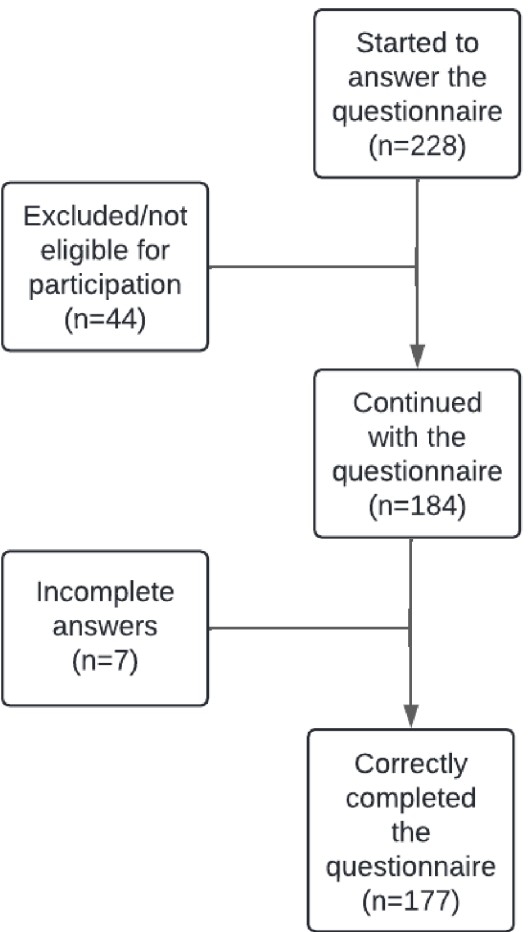

**Fig 1. Flow chart of the participants.**

friends, 11% dated different people, 7.9% dated only one person without any type of commitment, 43% maintained a stable relationship with one person, and 2.6% had a committed relationship. The mean age of the first relationship was 16.43 years (SD ± 2.06), and the mean number of partners per person at the time of the survey was 3.21 (SD ± 2.52). The mean duration of the relationships of those who had a partner at the time of questionnaire completion was 7.10 months (SD ±11.41), and the mean duration of the relationships was 3 months.

### 3.2. Cultural adaptation

No changes were made from the original version after back translation and discussion within the consensus groups.

The final Greek-translated version of the CADRI is shown in the S1 File.

### 3.3. Internal consistency

The internal consistency and reliability were good (>0.7), with Cronbach's alpha of 0.889 for the violence perpetrated subscale and 0.925 for the violence suffered subscale. The values obtained for each factor are shown in Table 1.

**Table 1. Internal consistency of CADRI factors.**

| Subscale | Factor | Items | Cronbach's Alpha (α) | McDonald's Omega (Ω) |
|---|---|---|---|---|
| Perpetration (α = 0.889) | Emotional and verbal violence | 4.A, 7.A, 9.A, 12.A, 17.A, 21.A, 23.A, 24.A, 28.A, 32.A | 0.865 | 0.868 |
| | Sexual violence | 2.A, 13.A, 15.A, 19.A | 0.585 | 0.664 |
| | Physical Violence | 8.A, 25.A, 30.A, 34.A | 0.803 | 0.808 |
| | Threats | 5.A, 29.A, 31.A, 33.A | 0.647 | 0.636 |
| | Relational Violence | 3.A, 20.A, 35.A | 0.535 | 0.538 |
| Victimization (α = 0.925) | Emotional and verbal violence | 4.B, 7.B, 9.B, 12.B, 17.B, 21.B, 23.B, 24.B, 28.B, 32.B | 0.878 | 0.881 |
| | Sexual violence | 2.B, 13.B, 15.B, 19.B | 0.824 | 0.853 |
| | Physical Violence | 8.B, 25.B, 30.B, 34.B | 0.827 | 0.829 |
| | Threats | 5.B, 29.B, 31.B, 33.B | 0.830 | 0.841 |
| | Relational Violence | 3.B, 20.B, 35.B | 0.853 | 0.854 |

The correlation coefficients between each item and the total scale score are shown in Table 2, with mostly high and positive correlations (r > 0.30) and some weak correlations (r > 0.10) for Items 2, 15, 19, and 35 of the perpetrated violence subscale.

### 3.4. Exploratory factor analysis

For the factor analysis, maximum likelihood analysis was used for the two subscales that make up the CADRI; in both cases, the extraction was forced to 5 factors to maintain the structure of the original instrument, and preliminary tests were carried out to ensure the adequacy of the data. For the perpetration subscale, 5 factors explained 50.46% of the total variance, resulting in an index of 0.84 in the KMO test for sampling adequacy and a significant Bartlett's test of sphericity ($X^2$ (300) = 2026.470; p < 0.001). For the victimization subscale, 5 factors explained 58.81% of the total variance; an index of 0.88 was obtained in the KMO test for sampling adequacy, and Bartlett's test of sphericity was significant ($X^2$ (300) = 2712.882; p < 0.001). A varimax rotation with Kaiser normalization was subsequently applied to facilitate interpretation; the matrix of factor loadings after varimax rotation for the victimization subscale is presented in Table 3 because it obtained the best adequacy.

The resulting factor structure is similar to that of the original instrument except for Items 21 and 33, where they should be assigned to the next factor with more weight.

### 3.5. Pilot study of the CADRI Greek version

The mean total score for the violence perpetration subscale was 8.52 (DS ± 8.49), and that for the victimization subscale was 10.23 (DS ± 11.31). It was found that it is common for couples to engage in violent acts even if they do not occur with high frequency. It was found that 88.7% of university students perpetrated DV in the last year on their partners, while 90.7% suffered DV. Table 4 shows the mean of the scores obtained for each type of violence, the standard deviation, and the percentage of the population sample affected by each type of violence disaggregated by sex, with verbal and emotional violence being the most common type of violence.

Table 5 shows the correlations obtained between the types of violence, revealing strongly significant relationships between violence perpetrated and victimization.

**Table 2. Interitem and total-item correlation matrix.**

**Perpetration subscale**

| Item | 2 | 3 | 4 | 5 | 7 | 8 | 9 | 12 | 13 | 15 | 17 | 19 | 20 | 21 | 23 | 24 | 25 | 28 | 29 | 30 | 31 | 32 | 33 | 34 | 35 | total |
|---|---|---|---|---|---|---|---|---|---|---|---|---|---|---|---|---|---|---|---|---|---|---|---|---|---|---|
| 2 | 1 | | | | | | | | | | | | | | | | | | | | | | | | | 0.176 |
| 3 | 0.181 | 1 | | | | | | | | | | | | | | | | | | | | | | | | 0.342 |
| 4 | 0.029 | 0.092 | 1 | | | | | | | | | | | | | | | | | | | | | | | 0.506 |
| 5 | 0.100 | 0.301 | 0.246 | 1 | | | | | | | | | | | | | | | | | | | | | | 0.458 |
| 7 | −0.035 | 0.090 | 0.352 | 0.253 | 1 | | | | | | | | | | | | | | | | | | | | | 0.557 |
| 8 | 0.089 | 0.262 | 0.271 | 0.500 | 0.251 | 1 | | | | | | | | | | | | | | | | | | | | 0..525 |
| 9 | 0.001 | 0.160 | 0.447 | 0.290 | 0.450 | 0.324 | 1 | | | | | | | | | | | | | | | | | | | 0.628 |
| 12 | 0.154 | 0.057 | 0.381 | 0.281 | 0.547 | 0.371 | 0.517 | 1 | | | | | | | | | | | | | | | | | | 0.656 |
| 13 | 0.250 | 0.204 | 0.214 | 0.121 | 0.121 | 0.356 | 0.204 | 0.210 | 1 | | | | | | | | | | | | | | | | | 0.379 |
| 15 | 0.309 | 0.437 | 0.101 | 0.128 | −0.069 | 0.086 | 0.104 | −0.042 | 0.331 | 1 | | | | | | | | | | | | | | | | 0.266 |
| 17 | 0.083 | 0.152 | 0.208 | 0.475 | 0.482 | 0.451 | 0.502 | 0.594 | 0.194 | 0.005 | 1 | | | | | | | | | | | | | | | 0.631 |
| 19 | 0.427 | 0.302 | 0.016 | 0.109 | −0.010 | 0.138 | 0.130 | 0.115 | 0.295 | 0.475 | 0.040 | 1 | | | | | | | | | | | | | | 0.185 |
| 20 | 0.065 | 0.533 | 0.122 | 0.333 | 0.231 | 0.440 | 0.203 | 0.165 | 0.086 | 0.157 | 0.331 | 0.101 | 1 | | | | | | | | | | | | | 0.451 |
| 21 | 0.112 | 0.416 | 0.176 | 0.317 | 0.208 | 0.423 | 0.254 | 0.378 | 0.352 | 0.199 | 0.465 | 0.089 | 0.432 | 1 | | | | | | | | | | | | 0.493 |
| 23 | 0.115 | 0.312 | 0.520 | 0.142 | 0.346 | 0.319 | 0.466 | 0.393 | 0.249 | 0.126 | 0.254 | 0.053 | 0.430 | 0.283 | 1 | | | | | | | | | | | 0.591 |
| 24 | 0.031 | 0.169 | 0.383 | 0.258 | 0.506 | 0.272 | 0.463 | 0.491 | 0.262 | 0.107 | 0.531 | 0.125 | 0.212 | 0.277 | 0.372 | 1 | | | | | | | | | | 0.625 |
| 25 | 0.287 | 0.288 | 0.318 | 0.304 | 0.258 | 0.620 | 0.343 | 0.426 | 0.475 | 0.272 | 0.443 | 0.120 | 0.480 | 0.580 | 0.464 | 0.352 | 1 | | | | | | | | | 0.653 |
| 28 | 0.038 | 0.092 | 0.368 | 0.110 | 0.467 | 0.193 | 0.354 | 0.379 | 0.179 | 0.177 | 0.213 | 0.041 | 0.201 | 0.084 | 0.435 | 0.390 | 0.300 | 1 | | | | | | | | 0.508 |
| 29 | 0.029 | 0.416 | 0.321 | 0.243 | 0.263 | 0.248 | 0.358 | 0.270 | 0.310 | 0.320 | 0.339 | 0.212 | 0.334 | 0.460 | 0.420 | 0.354 | 0.302 | 0.314 | 1 | | | | | | | 0.516 |
| 30 | 0.173 | 0.207 | 0.374 | 0.244 | 0.275 | 0.390 | 0.472 | 0.417 | 0.152 | 0.259 | 0.284 | 0.135 | 0.352 | 0.343 | 0.485 | 0.276 | 0.571 | 0.326 | 0.228 | 1 | | | | | | 0.581 |
| 31 | 0.079 | 0.333 | 0.211 | 0.368 | 0.241 | 0.058 | 0.365 | 0.321 | 0.151 | 0.254 | 0.410 | 0.079 | 0.296 | 0.307 | 0.359 | 0.285 | 0.268 | 0.145 | 0.380 | 0.413 | 1 | | | | | 0.486 |
| 32 | 0.137 | 0.094 | 0.276 | 0.359 | 0.425 | 0.290 | 0.330 | 0.483 | 0.098 | 0.094 | 0.487 | 0.067 | 0.224 | 0.271 | 0.291 | 0.512 | 0.405 | 0.491 | 0.282 | 0.386 | 0.403 | 1 | | | | 0.603 |
| 33 | 0.179 | 0.146 | 0.229 | 0.507 | 0.218 | 0.528 | 0.310 | 0.358 | 0.206 | 0.260 | 0.479 | 0.232 | 0.356 | 0.453 | 0.214 | 0.290 | 0.532 | 0.144 | 0.213 | 0.465 | 0.417 | 0.436 | 1 | | | 0.546 |
| 34 | 0.233 | 0.177 | 0.282 | 0.373 | 0.245 | 0.404 | 0.311 | 0.335 | 0.316 | 0.233 | 0.411 | 0.099 | 0.358 | 0.324 | 0.376 | 0.314 | 0.672 | 0.293 | 0.209 | 0.446 | 0.365 | 0.459 | 0.567 | 1 | | 0.569 |
| 35 | 0.132 | 0.336 | 0.200 | 0.077 | −0.019 | 0.163 | 0.244 | 0.052 | 0.301 | 0.537 | 0.106 | 0.232 | 0.250 | 0.116 | 0.270 | 0.147 | 0.232 | 0.263 | 0.239 | 0.177 | 0.176 | 0.079 | 0.182 | 0.185 | 1 | 0.285 |

**Victimization subscale**

| Item | 2 | 3 | 4 | 5 | 7 | 8 | 9 | 12 | 13 | 15 | 17 | 19 | 20 | 21 | 23 | 24 | 25 | 28 | 29 | 30 | 31 | 32 | 33 | 34 | 35 | total |
|---|---|---|---|---|---|---|---|---|---|---|---|---|---|---|---|---|---|---|---|---|---|---|---|---|---|---|
| 2 | 1 | | | | | | | | | | | | | | | | | | | | | | | | | 0.542 |
| 3 | 0.557 | 1 | | | | | | | | | | | | | | | | | | | | | | | | 0.674 |
| 4 | 0.146 | 0.250 | 1 | | | | | | | | | | | | | | | | | | | | | | | 0.394 |
| 5 | 0.300 | 0.461 | 0.258 | 1 | | | | | | | | | | | | | | | | | | | | | | 0.531 |
| 7 | 0.209 | 0.286 | 0.253 | 0.205 | 1 | | | | | | | | | | | | | | | | | | | | | 0.518 |
| 8 | 0.174 | 0.333 | 0.366 | 0.480 | 0.244 | 1 | | | | | | | | | | | | | | | | | | | | 0.563 |
| 9 | 0.310 | 0.450 | 0.409 | 0.276 | 0.429 | 0.335 | 1 | | | | | | | | | | | | | | | | | | | 0.636 |
| 12 | 0.376 | 0.393 | 0.245 | 0.407 | 0.507 | 0.376 | 0.512 | 1 | | | | | | | | | | | | | | | | | | 0.689 |
| 13 | 0.668 | 0.594 | 0.203 | 0.315 | 0.257 | 0.261 | 0.458 | 0.413 | 1 | | | | | | | | | | | | | | | | | 0.637 |
| 15 | 0.530 | 0.566 | 0.188 | 0.420 | 0.318 | 0.148 | 0.391 | 0.314 | 0.672 | 1 | | | | | | | | | | | | | | | | 0.584 |

*(Continued)*

**Table 2.** (Continued)

**Perpetration subscale**

| | 17 | 19 | 20 | 21 | 23 | 24 | 25 | 28 | 29 | 30 | 31 | 32 | 33 | 34 | 35 | total |
|----|----|----|----|----|----|----|----|----|----|----|----|----|----|----|----|-------|
| 17 | 1 | | | | | | | | | | | | | | | 0.667 |
| 19 | 0.349 | 1 | | | | | | | | | | | | | | 0.530 |
| 20 | 0.178 | 0.385 | 1 | | | | | | | | | | | | | 0.534 |
| 21 | 0.456 | 0.470 | 0.439 | 1 | | | | | | | | | | | | 0.575 |
| 23 | 0.408 | 0.473 | 0.474 | 0.400 | 1 | | | | | | | | | | | 0.682 |
| 24 | 0.555 | 0.304 | 0.303 | 0.436 | 0.430 | 1 | | | | | | | | | | 0.605 |
| 25 | 0.391 | 0.239 | 0.218 | 0.314 | 0.304 | 0.222 | 1 | | | | | | | | | 0.550 |
| 28 | 0.380 | 0.334 | 0.269 | 0.380 | 0.334 | 0.632 | 0.495 | 1 | | | | | | | | 0.623 |
| 29 | 0.382 | 0.313 | 0.251 | 0.293 | 0.382 | 0.409 | 0.247 | 0.364 | 1 | | | | | | | 0.553 |
| 30 | 0.388 | 0.380 | 0.187 | 0.249 | 0.196 | 0.228 | 0.700 | 0.236 | 0.270 | 1 | | | | | | 0.476 |
| 31 | 0.496 | 0.373 | 0.327 | 0.277 | 0.355 | 0.247 | 0.423 | 0.659 | 0.277 | 0.381 | 1 | | | | | 0.632 |
| 32 | 0.418 | 0.104 | 0.303 | 0.265 | 0.406 | 0.228 | 0.270 | 0.430 | 0.463 | 0.170 | 0.411 | 1 | | | | 0.514 |
| 33 | 0.377 | 0.313 | 0.346 | 0.322 | 0.289 | 0.247 | 0.428 | 0.282 | 0.463 | 0.608 | 0.545 | 0.355 | 1 | | | 0.546 |
| 34 | 0.488 | 0.201 | 0.344 | 0.252 | 0.186 | 0.354 | 0.621 | 0.638 | 0.479 | 0.401 | 0.638 | 0.346 | 0.684 | 1 | | 0.547 |
| 35 | 0.210 | 0.396 | 0.712 | 0.391 | 0.209 | 0.347 | 0.299 | 0.184 | 0.375 | 0.327 | 0.310 | 0.257 | 0.153 | 0.257 | 1 | 0.511 |

**Table 3. Map of item factor loadings.**

**Dating Violence Victimization**

| Items | Factor 1 | Factor 2 | Factor 3 | Factor 4 | Factor 5 |
|---|---|---|---|---|---|
| 24.B He or She blamed me for the problem. | **0.737** | 0.169 | 0.142 | 0.012 | 0.120 |
| 12.B He or She spoke to me in a hostile or malicious tone of voice. | **0.680** | 0.205 | 0.233 | 0.257 | 0.007 |
| 7.B He or She mentioned something bad I had done in the past. | **0.649** | 0.066 | 0.106 | 0.105 | 0.074 |
| 28.B He or She accused me of flirting with another boy/girl. | **0.614** | 0.241 | 0.137 | 0.097 | 0.149 |
| 9.B He or She said things to make me angry. | **0.614** | 0.291 | 0.090 | 0.120 | 0.125 |
| 17.B He or She insulted me with derogatory expressions or words. | **0.585** | 0.257 | 0.319 | 0.263 | -0.057 |
| 23.B He or She was watching who I was with and where I was. | **0.552** | 0.346 | 0.026 | 0.219 | 0.283 |
| 32.B He or She threatened to end the relationship. | **0.535** | -0.062 | 0.109 | 0.375 | 0.164 |
| 4.B He or She did something to make me jealous. | **0.319** | 0.095 | 0.216 | 0.121 | 0.041 |
| 21.B He or She made fun of me or mocked me in front of others. | **0.357**[*] | **0.447** | 0.260 | -0.008 | 0.267 |
| 13.B He or She made me have sex when I did not want to have sex. | 0.282 | **0.909** | 0.051 | 0.105 | 0.097 |
| 19.B He or She kissed me when I did not want him to. | 0.184 | **0.700** | 0.076 | 0.132 | 0.190 |
| 2.B He or She touched me sexually when I did not want him or her to. | 0.150 | **0.640** | 0.206 | 0.126 | 0.238 |
| 15.B He or She threatened me in an attempt to have sex with me. | 0.180 | **0.629** | 0.219 | 0.237 | 0.100 |
| 30.B He or She slapped me or pulled my hair. | 0.128 | 0.132 | **0.836** | 0.055 | 0.108 |
| 34.B He or She pushed me or shook me. | 0.239 | 0.118 | **0.725** | 0.285 | 0.019 |
| 25.B He or She kicked, hit or punched me. | 0.188 | 0.186 | **0.707** | 0.185 | 0.071 |
| 8.B He or She threw something (an object) at me. | 0.271 | 0.111 | **0.482** | 0.372 | 0.071 |
| 33.B He or She threatened to hit me or throw an object at me. | 0.156 | 0.087 | **0.557** | **0.465**[*] | 0.189 |
| 31.B He or She threatened to hurt me. | 0.178 | 0.275 | 0.301 | **0.745** | 0.114 |
| 29.B He or She deliberately tried to scare me. | 0.265 | 0.125 | 0.177 | **0.661** | 0.119 |
| 5.B He or She destroyed or threatened to destroy something of value to me. | 0.165 | 0.199 | 0.226 | **0.587** | 0.173 |
| 20.B He or She said things to my friends about me to turn them against me. | 0.189 | 0.208 | 0.105 | 0.131 | **0.944** |
| 35.B He or She spread rumors about me. | 0.188 | 0.351 | 0.086 | 0.229 | **0.612** |
| 3.B He or She tried to turn my friends against me. | 0.258 | 0.489 | 0.182 | 0.211 | **0.560** |

**Table 4. Types of dating violence obtained with the GREEK version of the CADRI (n = 177).**

| Type of Violence | Men | | | Women | | | Total | | |
|---|---|---|---|---|---|---|---|---|---|
| | **M** | **SD ±** | **%** | **M** | **SD ±** | **%** | **M** | **SD ±** | **%** |
| Verbal-emotional Perpetration | 5,07 | 4.21 | 86.2% | 7.29 | 6.34 | 86.7% | 6.91 | 6.08 | 86.6% |
| Verbal-emotional Victimization | 6.57 | 6.20 | 90% | 7.36 | 6.67 | 90.3% | 7.22 | 6.58 | 90.2% |
| Sexual Perpetration | 0.65 | 1.52 | 24.1% | 0.25 | 0.76 | 15.6% | 0.32 | 0.93 | 17.0% |
| Sexual Victimization | 0.63 | 1.56 | 23.3% | 0.99 | 2.15 | 29.3% | 0.93 | 2.06 | 28.2% |
| Physical Perpetration | 0.33 | 1.29 | 13.3% | 0.47 | 1.27 | 19.9% | 0.45 | 1.27 | 18.8% |
| Physical Victimization | 1.00 | 2.27 | 23.3% | 0.48 | 1.45 | 17.0% | 0.58 | 1.63 | 18.1% |
| Threats Perpetration | 0.50 | 1.28 | 20.0% | 0.48 | 1.22 | 19.9% | 0.49 | 1.22 | 19.9% |
| Threats Victimization | 1.07 | 2.26 | 33.3% | 0.81 | 2.00 | 24.1% | 0.86 | 2.04 | 25.7% |
| Relational Perpetration | 0.37 | 1.03 | 13.3% | 0.18 | 0.64 | 10.3% | 0.21 | 0.72 | 10.8% |
| Relational Victimization | 0.97 | 1.71 | 26.7% | 0.69 | 1.76 | 21.1% | 0.74 | 1.75 | 22.0% |
| Total Perpetration | 8.21 | 7.73 | 89.3% | 9.79 | 8.64 | 88.6% | 8.52 | 8.49 | 88.7% |
| Total Victimization | 10.23 | 11.54 | 90% | 10.23 | 11.30 | 90.8% | 10.22 | 11.31 | 90.7% |

**Table 5. Correlations between violence types.**

| | Verbal emotional Perpetration | Verbal emotional Victimization | Sexual Perpetration | Sexual Victimization | Physical Perpetration | Physical Victimization | Threats Perpetration | Threats Victimization | Relational Perpetration | Relational Victimization |
|---|---|---|---|---|---|---|---|---|---|---|
| Verbal emotional Perpetration | 1 | | | | | | | | | |
| Verbal emotional Victimization | 0.756** | 1 | | | | | | | | |
| Sexual Perpetration | 0.041 | 0.050 | 1 | | | | | | | |
| Sexual Victimization | 0.201** | 0.195* | 0.384** | 1 | | | | | | |
| Physical Perpetration | 0.193* | 0.157* | 0.051 | 0.122 | 1 | | | | | |
| Physical Victimization | 0.140 | 0.150* | 0.177* | 0.332** | 0.491** | 1 | | | | |
| Threats Perpetration | 0.195* | 0.162* | 0.188* | 0.205** | 0.416** | 0.330** | 1 | | | |
| Threats Victimization | 0.197** | 0.193* | 0.181* | 0.252** | 0.296** | 0.446** | 0.570** | 1 | | |
| Relational Perpetration | 0.135 | 0.109 | 0.140 | 0.273** | 0.254** | 0.358** | 0.348** | 0.230** | 1 | |
| Relational Victimization | 0.168* | 0.171* | 0.048 | 0.340** | 0.137 | 0.459** | 0.119 | 0.419** | 0.387** | 1 |

**The correlation is significant at 0.01 (bilateral).

*The correlation is significant at 0.05 (bilateral).

## 4. Discussion

According to the results, the overall internal validity of the Greek version of the scale is quite good, except for the subscale related to the perpetration of sexual violence, threats, and relational violence. These values were similar to those obtained in the original scale [20] and the Spanish version [22]. Although the results obtained were better for the violence victimization subscale, we could not compare the data with the original data, since it addresses only the internal consistency of the perpetrated subscale. However, we could compare it with the Spanish version, where, in contrast to our results, Cronbach's alpha was less than 0.7 for sexual violence and threats.

The correlation coefficients between each item and the total scale score showed mostly high and positive correlations (r > 0.30), suggesting that most of the items were well aligned with the total scale and were representative of the construct being measured [33].

For the EFA, the structure was indeed forced to 5 factors to maintain the structure of the original scale and to be able to compare the obtained results with those of other studies correctly. However, among the different versions of the scale, we found significant changes in its structure, as in the case of the validation in the Colombian population [23], where the physical violence factor was eliminated. Additionally, the factor analysis was carried out in the university population [26], where the sexual violence factor was eliminated, and the validation was carried out in the young population [27], where we found six factors in total between the two subscales of violence. The short versions also affect the factor structure [24,25].

The factor structure we found in our results is quite similar to the structure proposed in the original scale except for two items, which have high factor loadings in more than one factor and, thus, were reallocated to the original scale factor. This result was similar to that obtained by Fernandez-Fuertes [22] in the validation of the scale in Spanish, where some of the items of the threat factor were not adequately distributed. However, the original structure proposed by Wolfe et al. [20] was maintained. The data obtained from the Greek population revealed a high incidence of violence, especially verbal and emotional violence. Importantly, CADRI

assesses not only the frequency of violent acts in relationships but also whether these acts occur. Special consideration should be given to the age of the participants in the study and the short duration of the relationships [32].

The prevalence rates of violence are high, although they are in line with those reported in other studies [34] and even with some results at the European level [2]. A few years ago, some Spanish studies warned of the high rates of psychological or verbal/emotional violence, victimization and perpetration associated with the use of CADRIs among the young population, with no significant differences according to sex [31,35–38].

Despite the negative consequences for the well-being of being involved in DV [39], there is high tolerance and acceptance of this type of violence related to benevolent sexism [40]. It is common to find acceptance of behaviors such as control and manipulative attitudes hidden behind love or concern, such as talking about toxic or unhealthy relationships [7,41,42]. However, owing to the results obtained, it is necessary to implement awareness and prevention campaigns to reduce violence at this stage, since it has been shown that DV is related to suffering and the perpetration of violence in the future [43,44].

Acceptance of violence is related not only to being a victim of DV but also to the perpetration of DV [45]. As shown in the results, there is a strong correlation between perpetrating and suffering from all types of violence; several studies indicate that not only does this correlation exist but also that people who suffer violence are more likely to commit and justify the use of violence [43,46–48]. Furthermore, according to a meta-analysis, there is a relationship between the type of attachment in the couple and the acceptance and execution of violent behaviors, such as anxious attachment, avoidant attachment, and disorganized attachment styles [49].

DV is a public health problem that should continue to be studied in the Greek population since these students will be future nurses and will have to care for people who suffer from the same problem. They must know how to recognize and help people who suffer from DV since health services are the entry point for possible victims of violence.

## 4.1. Limitations

Despite the efforts made to ensure the validity and reliability of our results, this study has several limitations that should be considered when the findings are interpreted. First, as this is a self-answer questionnaire, participants may be dishonest in their responses. Additionally, there were a greater number of women than men in the sample, although no significant differences by gender were found because most students in the nursing program are women. Finally, the descriptive study is a pilot study, and the results cannot be generalized due to the lack of a representative sample. Future research could address these limitations to strengthen the validity and applicability of the questionnaire in diverse contexts.

As strengths, the double back-translation and expert evaluation of the Greek version of the questionnaire before distribution to the students should be highlighted. We also obtained the necessary sample to validate the scale, which is composed of 25 items on violence and 10 items on positive behaviors, which have not been evaluated in any previous version.

## 5. Conclusion

In conclusion, the results of this study provide robust evidence of the validity and reliability of the CADRI for measuring DV in the Greek university population.

CADRI's adaptation to Greek is a breakthrough for the study of DV in the Greek population, which until now lacked culturally adapted and validated tools to measure violence in their mother tongue.

Due to the high prevalence of DV in general, especially in verbal and emotional violence, research on DV in this population is necessary to create useful prevention strategies adapted to victims and perpetrators of DV. Bearing in mind that the university is where future health and education professionals are trained, they should know how to recognize, avoid, and help prevent cases of DV.

## Supporting information

**S1 File. Conflict in adolescent dating relationships (CADRI) Greek version.**
(PDF)

## Author contributions

**Conceptualization:** Esperanza Barroso-Corroto, Juan Manuel Carmona-Torres, Michail Zografakis-Sfakianakis.

**Data curation:** Esperanza Barroso-Corroto, Evangelia Kartsoni.

**Formal analysis:** Esperanza Barroso-Corroto, Juan Manuel Carmona-Torres, Antonios Christodoulakis.

**Funding acquisition:** Juan Manuel Carmona-Torres.

**Investigation:** Esperanza Barroso-Corroto.

**Methodology:** Esperanza Barroso-Corroto, Juan Manuel Carmona-Torres, Evangelia Kartsoni.

**Project administration:** Juan Manuel Carmona-Torres.

**Resources:** Esperanza Barroso-Corroto, Evangelia Kartsoni.

**Supervision:** Juan Manuel Carmona-Torres, José Alberto Laredo-Aguilera, Athina Patelarou, Michail Zografakis-Sfakianakis.

**Validation:** Esperanza Barroso-Corroto, Evangelia Kartsoni.

**Visualization:** Ángel López-Fernández-Roldan, Carlos Navarrete-Tejero.

**Writing – original draft:** Esperanza Barroso-Corroto, Michail Zografakis-Sfakianakis.

**Writing – review & editing:** Juan Manuel Carmona-Torres, José Alberto Laredo-Aguilera, Ángel López-Fernández-Roldan, Carlos Navarrete-Tejero, Antonios Christodoulakis, Athina Patelarou, Michail Zografakis-Sfakianakis.

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
