## [Decision Letter · Decision Letter 0]

29 Nov 2024

PONE-D-24-46432Cultural adaptation, translation and validation of conflict in adolescence dating relationship inventory (CADRI) to Greek languagePLOS ONE

Dear Dr. Carmona-Torres,

Thank you for submitting your manuscript to PLOS ONE. After careful consideration, we feel that it has merit but does not fully meet PLOS ONE’s publication criteria as it currently stands. Therefore, we invite you to submit a revised version of the manuscript that addresses the points raised during the review process. 

We look forward to receiving your revised manuscript.

Kind regards,

Marianna Mazza

Academic Editor

PLOS ONE

Journal Requirements:

2. Thank you for stating the following financial disclosure: This research was funded by a grant from the European Regional Development Fund (ERDF) [Fondo Europeo de Desarrollo Regional (FEDER), (DOCM 27/01/2021)].

Esperanza Barroso-Corroto is supported by a grant (SBPLY/23/180502/000002) from the Junta de Comunidades de Castilla-La Mancha (Spain) and cofnanced by the Fondo Social Europeo Plus (ESF+ 2021-2027) Program. 

3. In the online submission form, you indicated that the results of this research are published in this article. The data that support the findings of this study are available from the corresponding author upon reasonable request.

Reviewers' comments:

Reviewer's Responses to Questions

**Comments to the Author**

1. Is the manuscript technically sound, and do the data support the conclusions?

Reviewer #1: Partly

Reviewer #2: Yes

2. Has the statistical analysis been performed appropriately and rigorously? 

Reviewer #1: No

Reviewer #2: Yes

3. Have the authors made all data underlying the findings in their manuscript fully available?

Reviewer #1: Yes

Reviewer #2: Yes

4. Is the manuscript presented in an intelligible fashion and written in standard English?

Reviewer #1: Yes

Reviewer #2: Yes

5. Review Comments to the Author

Reviewer #1: I have divided comments by section.

Abstract:

- Sentence 1: While sentence 1 is grammatically sound, it reads a bit awkward. Consider rephrasing such that the sentence does not end with “it.”

- Unclear in the methods if there was expert review of the instrument conducted, please specify.

- Typo, Methods section should say “methods” not “method”

Introduction

- In the second paragraph, you discuss descriptive statistics of dating violence. What age group are these estimates for?

- In the second paragraph, you note that these data are consistent with North American estimates, though given the wide range of estimates, I feel this statement is a bit superfluous.

- In the third paragraph you write that alcohol consumption, marijuana use, and prior exposure to domestic violence is “related” to dating violence. What does “related” mean? Please be more specific.

- What defines young adult and adolescent population…what age ranges?

- You write: “Intimate partner violence (IPV) in the Greek adult population has been little studied (17,18) and they use the Conflict Tactics Scale (CTS2) to assess the prevalence of IPV by performing multicentre studies by doing a careful back-translation, as the previous ones.” Who is “they”? Please reword this section to be more clear.

- You write: “However, according to the systematic review (20) where 29 instruments for measuring DV were evaluated and CADRI was the most recommended one for measuring DV.” What does “most recommended” mean in the context of this study? Additionally, see general comments for my grammatical errors.

Methods

- Section 2.1 typo should say “design”

- Please expand Design section. Specifically, discuss the steps of your scale development in greater detail. You mention a few large steps: 1) translation, cross-cultural item adaption, and content validation 2) face validation 3) piloting on a small sample, validity, and reliability.

- You mention that you measured convergent validity—where is this discussed?

- You mention inclusion criteria, but how were participants recruited for this survey?

- Consider using the term “Likert-like or Likert-type scale” as Likert scales traditionally feature five or seven items, not four.

- Who conducted the backward and forward translation of the preliminary Greek version? Please avoid passive voice.

- In the last paragraph of the Procedure subsection, you write that the questionnaire was distributed to students during academic lectures. This raises a few methodological concerns: do all students attend lectures? Were students’ grades affected by whether students took this questionnaire or not? Have you considered any desirability and sample biases?

- You mention the use of McDonald’s omega, please justify why. McDonald’s m

General Comments:

1) The goal of this project was to translate and culturally adapt the CADRI to the Greek population. I’m a bit confused because in the conclusion, you mention that you provided robust evidence of the validity and reliability of CADRI for Greek university students specifically. Please advise.

2) I would consider some cognitive interviewing specifically with regards to adapting this scale to Greek contexts.

3) There are grammatical issues specifically with tense in this paper; sometimes the writers write in present tense and then switch to past tense. Please address.

4) Please address typological errors in this paper. For instance, under sub-section “procedure,” you write “liker” instead of “Likert.”

5) At times, the writing style of this paper is not grammatically sound and/or does not read clearly. For instance, in the introduction, you write “However, according to the systematic review (20) where 29 instruments for measuring DV were evaluated and CADRI was the most recommended one for measuring DV.” Here, the use of “however” indicates there will be an independent clause after it but what follows it is dependent. This sentence is also a run-on sentence and the mention of “measuring DV” twice makes this a bit awkward sounding.

Reviewer #2: The way the paper is written is too periphrastic and wordy. Sometimes the words used to describe a concept are not apt. WHat's liker? Did you mean Likert? In Data analysis, you have put a piece of your project or protocol (sphericity will be calculated); please addres this.

In sociodemographic Characteristics, what's the "undefined" gender? I'd prefer not to say? "2.2% claimed to have a different sexual orientation", what is a different orientation?

In internal consistency, I wouldn't call a Pearson's r>0.1 a moderate correlation.

In your factor analysis, the item 33B was significant for two factors. You held the stronger, you should have declared this before and set the value for belonging to a factor at 0.4.

The Tables have too many commas instead of points. Please correct.

The first paragraph of the Discussion is quite unclear.

The sentence: "...except for two items, which have high factor loadings on all the items", what do you mean? Which items, on how many factors did they load?

Why didn't you include other types of univerrsity students? Do you believe nurses are representative of the Greek youth? Do you believe Greek nurses are inclined to DV? The results obtained were very high.

6. PLOS authors have the option to publish the peer review history of their article (what does this mean? ). If published, this will include your full peer review and any attached files.

**Do you want your identity to be public for this peer review?** For information about this choice, including consent withdrawal, please see our Privacy Policy .

Reviewer #1: No

Reviewer #2: **Yes: ** Georgios Kotzalidis

---

## [Author Response · Author response to Decision Letter 0]

11 Dec 2024

We appreciate your constructive comments, helpful information, and your time. Thanks to this review, our manuscript has been substantially improved. Responses to your comments are bold and the changes in the manuscript have been tracked and highlighted in yellow.

Reviewer Comments:

Reviewer #1:

I have divided comments by section:

Abstract:

- Sentence 1: While sentence 1 is grammatically sound, it reads a bit awkward. Consider rephrasing such that the sentence does not end with “it.”

- Unclear in the methods if there was expert review of the instrument conducted, please specify.

- Typo, Methods section should say “methods” not “method”

Thank you very much for your comments. The requested changes have been incorporated in the Abstract. The first sentence has been modified “Dating violence (DV) is a major public health problem with serious consequences in the young population”. The expert review of the instrument has been added to the methodology section, and “method” has been changed to “methods”.

The manuscript has been edited in English since many proposals were to improve the language.

Introduction

- In the second paragraph, you discuss descriptive statistics of dating violence. What age group are these estimates for?

The systematic reviews cited in the text focus their study on adolescents, however, these results are similar to those found in studies with young adults, so we have modified the paragraph to a better understanding and added another reference.

“In Europe, studies on DV have been carried out in several countries such as Spain, Italy, the UK, and Germany. The prevalence rates of DV in Europe in adolescents according to a systematic review (2) varied for psychological victimization ranged from 5.9% to 95.5% for females and from 5.6% to 94.5% for males, physical DV victimization from 2.2% to 32.9% for females and from 0.8% to 29.8% for males, sexual DV victimization from 4.8% to 41.0% for female and from 2.4% to 39.0% for male. These data are consistent with those found in other studies conducted mainly in North America, and in other countries where this phenomenon has been studied (3), as well as those carried out in young adults (4).”

- In the second paragraph, you note that these data are consistent with North American estimates, though given the wide range of estimates, I feel this statement is a bit superfluous.

In the systematic reviews reviewed, most of the studies were carried out in the United States and concluded with similar findings and high variability of results. However, it is true that they also include studies from other countries such as Canada, Finland, Spain, or Colombia.

- In the third paragraph you write that alcohol consumption, marijuana use, and prior exposure to domestic violence is “related” to dating violence. What does “related” mean? Please be more specific.

“Related” refers to other associated factors, since it has changed in the text for better understanding.

“In addition, it has been observed that alcohol consumption, marijuana use, having been exposed to domestic violence, and participation in vandalism are other factors associated with DV (10,11).”

- What defines young adult and adolescent population…what age ranges?

According to the Cambridge Dictionary, a “young adult” is a person who is in his or her late teens or early twenties. Also, the WHO and UN define “young adults” as a period between 15 and 24 years of age.

- You write: “Intimate partner violence (IPV) in the Greek adult population has been little studied (17,18) and they use the Conflict Tactics Scale (CTS2) to assess the prevalence of IPV by performing multicentre studies by doing a careful back-translation, as the previous ones.” Who is “they”? Please reword this section to be more clear.

To clarify, the sentence form has been changed to the passive form

“Intimate partner violence (IPV) in the Greek adult population has been little studied (18,19). Although there are few studies, in these studies the Conflict Tactics Scale (CTS2) has been used to assess the prevalence of IPV. Also, in these studies, the researchers used a careful back-translation without instrument validation. There are no instruments to measure DV or IPV adapted, translated, and validated in Greek, making it difficult to study DV in this population.”

- You write: “However, according to the systematic review (20) where 29 instruments for measuring DV were evaluated and CADRI was the most recommended one for measuring DV.” What does “most recommended” mean in the context of this study? Additionally, see general comments for my grammatical errors.

The review refers to it as the most suitable instrument for measuring DV. In the text, “recommended” has been changed to “suitable”.

Methods

- Section 2.1 typo should say “design”

Thank you for your comments. The change has been made.

- Please expand Design section. Specifically, discuss the steps of your scale development in greater detail. You mention a few large steps: 1) translation, cross-cultural item adaption, and content validation 2) face validation 3) piloting on a small sample, validity, and reliability.

Further details are provided in section 2.4. Procedure

- You mention that you measured convergent validity—where is this discussed?

Finally, convergent validity could not be carried out due to the lack of validated instruments measuring violence in the Greek context. The text has been corrected and excess information has been removed.

- You mention inclusion criteria, but how were participants recruited for this survey?

The following information has been added to the manuscript:

To reach the sample size participants were recruited during academic classes to ensure as much participation as possible. A member of the research team asked the teachers for permission, gave the study information and requested the students' voluntary participation. Once accepted to participate, the informed consent form was given to the students for their signature.

- Consider using the term “Likert-like or Likert-type scale” as Likert scales traditionally feature five or seven items, not four.

Thank you very much for your assistance. We have change it.

- Who conducted the backward and forward translation of the preliminary Greek version? Please avoid passive voice.

One of the translators was Evangelia Kartsoni, she is Greek bilingual and an expert in the English language, and the other was Michail Zografakis Sfakianakis. After back-translation of the scale by both authors, the best version of each item was decided by a group of experts to ensure the accuracy of the scale.

The passive voice has been avoided from the text.

- In the last paragraph of the Procedure subsection, you write that the questionnaire was distributed to students during academic lectures. This raises a few methodological concerns: do all students attend lectures? Were students’ grades affected by whether students took this questionnaire or not? Have you considered any desirability and sample biases?

The questionnaire was distributed in different lectures, and imparted in different courses, some of them of compulsory attendance. Students were also asked to distribute the survey among those who did not attend.

Before the dissemination of the questionnaire, the professors agreed that collaboration and participation in the study were not detrimental to the academic performance of the participants.

Attempts were made to reduce sample bias by reaching out to as many different lectures as possible and providing clear information about the study, pointing out the importance of the participation of all of them.

- You mention the use of McDonald’s omega, please justify why. McDonald’s m

Although Cronbach's Alpha met the expected assumptions, it was considered appropriate to also measure McDonald's Omega due to the heterogeneity of the items and since it is a scale with multiple factors.

Hayes AF, Coutts JJ. Use Omega Rather than Cronbach’s Alpha for Estimating Reliability. But…. Commun Methods Meas. 2020 Jan 2;14(1):1–24. Available from: https://www.tandfonline.com/doi/abs/10.1080/19312458.2020.1718629

Roco-Videla Á, Aguilera-Eguía R, Olguin-Barraza M, Roco-Videla Á, Aguilera-Eguía R, Olguin-Barraza M. Ventajas del uso del coeficiente de omega de McDonald frente al alfa de Cronbach. Nutr Hosp. 2024 Jan 1;41(1):262–3. Available from: https://scielo.isciii.es/scielo.php?script=sci_arttext&pid=S0212-16112024000100030&lng=es&nrm=iso&tlng=es

General Comments:

1) The goal of this project was to translate and culturally adapt the CADRI to the Greek population. I’m a bit confused because in the conclusion, you mention that you provided robust evidence of the validity and reliability of CADRI for Greek university students specifically. Please advise.

The population is a central element in the validation of an instrument; therefore, it is necessary to specify the type of population where the study has been conducted to ensure the relevance, reliability, and replicability of the study. For this reason, the specific population of university students is noted in the conclusions.

For clarity, the objective has been modified as follows:

“this study aimed to culturally adapt, validate, and translate CADRI in the Greek language, to ensure its reliability and cultural and linguistic validity in the Greek population, specifically in young adults that constitute the university population.”

2) I would consider some cognitive interviewing specifically with regards to adapting this scale to Greek contexts.

Thank you for your contribution. The instrument has gone through a process of double backtranslation and face validation of the scale by the group of experts, to maintain the original meaning and comprehension of the items in the Greek context. In addition, students were asked about the correct understanding of the scale.

3) There are grammatical issues specifically with tense in this paper; sometimes the writers write in present tense and then switch to past tense. Please address.

Thank you very much for your comment, this has been considered and the verb tenses of the manuscript have been revised and modified. The text has been revised and edited in English.

4) Please address typological errors in this paper. For instance, under sub-section “procedure,” you write “liker” instead of “Likert.”

Following your recommendations, appropriate modifications have been made to improve the manuscript and edit it.

5) At times, the writing style of this paper is not grammatically sound and/or does not read clearly. For instance, in the introduction, you write “However, according to the systematic review (20) where 29 instruments for measuring DV were evaluated and CADRI was the most recommended one for measuring DV.” Here, the use of “however” indicates there will be an independent clause after it but what follows it is dependent. This sentence is also a run-on sentence and the mention of “measuring DV” twice makes this a bit awkward sounding.

Thank you for your recommendations, appropriate modifications have been made to improve the manuscript.

Reviewer #2:

The way the paper is written is too periphrastic and wordy. Sometimes the words used to describe a concept are not apt. WHat's liker? Did you mean Likert?

Thank you very much for your revisions, changes and grammatical corrections that have been made for a better understanding of the manuscript. Errors such as “liker” have been corrected and removed.

In Data analysis, you have put a piece of your project or protocol (sphericity will be calculated); please addres this.

Thank you very much for your input, the verb tense has been corrected.

In sociodemographic Characteristics, what's the "undefined" gender? I'd prefer not to say? "2.2% claimed to have a different sexual orientation", what is a different orientation?

In the field of gender identity, undefined gender relates to those who do not identify with traditional gender categories (male or female). This includes, but is not limited to:

- Non-binary people: identified outside of the binary gender spectrum.

- Gender fluid: People whose gender identity changes over time or according to circumstances.

Sexual orientation is an essential part of human identity. Many people explore and redefine their orientation throughout their lives. Moreover, these terms are not mutually exclusive, and people may identify with more than one category or with none. We usually ask only about the most common sexual orientations such as heterosexuality, bisexuality or homosexuality, ignoring others such as asexuality, antrosexuality, Skoliosexuality, Sapiosexuality, Omnisexuality or Pansexuality. Therefore, when asked about a different sexual orientation, we refer to these categories among others.

In internal consistency, I wouldn't call a Pearson's r>0.1 a moderate correlation.

Thanks for your contribution, the wording has been changed to weak.

In your factor analysis, the item 33B was significant for two factors. You held the stronger, you should have declared this before and set the value for belonging to a factor at 0.4.

The text explains “As can be observed, the resulting factor structure is similar to that of the original instrument except in items 21 and 33 where they should be assigned to the next factor with more weight.”

In the table, the factor with the highest weight is shown in bold, but the factor to which it belongs in the original scale is marked with an *.

In addition, the discussion debated the factorial structures of other validations of the same scale and defended maintaining the structure of the original scale.

The table has been modified to make this more understandable.

The Tables have too many commas instead of points. Please correct.

Thanks to your comment, the problem has been corrected.

The first paragraph of the Discussion is quite unclear.

An attempt has been made to improve the wording of the first paragraph to make it more understandable.

The sentence: "...except for two items, which have high factor loadings on all the items", what do you mean? Which items, on how many factors did they load?

Thank you very much for your contribution, the sentence has been changed to provide the missing information. As explained above, the original structure was maintained and the items with high factor weights in more than one factor were 21 and 33.

“The factor structure we found in our results is quite similar to the structure proposed in the original scale except for two items, which have high factor loadings in more than one factor and were reallocated to the original scale factor”

Why didn't you include other types of university students? Do you believe nurses are representative of the Greek youth? Do you believe Greek nurses are inclined to DV? The results obtained were very high.

Consent was only obtained from the ethics committee to carry out the study in the nursing faculty.

Nursing students are as representative of college students and young adults as students in any other faculty.

The fact that in Greece we do not find any validated instrument on intimate partner violence, nor studies carried out on any type of population is significant of the lack of interest in the subject of study and assimilation of the abuse by the population. Moreover, the high prevalence is found mainly in verbal and emotional violence which is the most socially accepted and normalized type of violence.

Other studies in different countries find similar prevalences with the same instrument, indicating that dating violence is a problem that needs to be addressed.

---

## [Decision Letter · Decision Letter 1]

6 Jan 2025

Cultural adaptation, translation and validation of the conflict in adolescence dating relationship inventory (CADRI) in the Greek language

PONE-D-24-46432R1

Dear Dr. Carmona-Torres,

We’re pleased to inform you that your manuscript has been judged scientifically suitable for publication and will be formally accepted for publication once it meets all outstanding technical requirements.

Kind regards,

Marianna Mazza

Academic Editor

PLOS ONE

Additional Editor Comments (optional):

Reviewers' comments:

Reviewer's Responses to Questions

**Comments to the Author**

1. If the authors have adequately addressed your comments raised in a previous round of review and you feel that this manuscript is now acceptable for publication, you may indicate that here to bypass the “Comments to the Author” section, enter your conflict of interest statement in the “Confidential to Editor” section, and submit your "Accept" recommendation.

Reviewer #2: All comments have been addressed

2. Is the manuscript technically sound, and do the data support the conclusions?

Reviewer #2: Yes

3. Has the statistical analysis been performed appropriately and rigorously? 

Reviewer #2: Yes

4. Have the authors made all data underlying the findings in their manuscript fully available?

Reviewer #2: Yes

5. Is the manuscript presented in an intelligible fashion and written in standard English?

Reviewer #2: Yes

6. Review Comments to the Author

Reviewer #2: Authors addressed my comments adequately. I thank authors for addressing my comments and wish their study could be published in Plos One.

7. PLOS authors have the option to publish the peer review history of their article (what does this mean? ). If published, this will include your full peer review and any attached files.

**Do you want your identity to be public for this peer review?** For information about this choice, including consent withdrawal, please see our Privacy Policy .

Reviewer #2: **Yes: ** Georgios D. Kotzalidis

---

## [Editor Report · Acceptance letter]

PONE-D-24-46432R1

PLOS ONE

Dear Dr. Carmona-Torres,

I'm pleased to inform you that your manuscript has been deemed suitable for publication in PLOS ONE. Congratulations! Your manuscript is now being handed over to our production team.

Kind regards,

on behalf of

Dr. Marianna Mazza

Academic Editor

PLOS ONE